# [^89^Zr]-Pertuzumab PET Imaging Reveals Paclitaxel Treatment Efficacy Is Positively Correlated with HER2 Expression in Human Breast Cancer Xenograft Mouse Models

**DOI:** 10.3390/molecules26061568

**Published:** 2021-03-12

**Authors:** Yun Lu, Meng Li, Adriana V. F. Massicano, Patrick N. Song, Ameer Mansur, Katherine A. Heinzman, Benjamin M. Larimer, Suzanne E. Lapi, Anna G. Sorace

**Affiliations:** 1Department of Radiology, University of Alabama at Birmingham, Birmingham, AL 35233, USA; yunlu@uab.edu (Y.L.); lily01@uab.edu (M.L.); amassicano@uabmc.edu (A.V.F.M.); psong@uabmc.edu (P.N.S.); bmlarimer@uabmc.edu (B.M.L.); lapi@uab.edu (S.E.L.); 2Graduate Biomedical Sciences, University of Alabama at Birmingham, Birmingham, AL 35233, USA; 3Department of Biomedical Engineering, University of Alabama at Birmingham, Birmingham, AL 35233, USA; amansur@uab.edu (A.M.); kheinz@uab.edu (K.A.H.); 4O’Neal Comprehensive Cancer Center, University of Alabama at Birmingham, Birmingham, AL 35233, USA; 5Department of Chemistry, University of Alabama at Birmingham, Birmingham, AL 35233, USA

**Keywords:** positron emission tomography, FDG-PET, heterogeneity, HER2+ breast cancer, molecular imaging

## Abstract

Paclitaxel (PTX) treatment efficacy varies in breast cancer, yet the underlying mechanism for variable response remains unclear. This study evaluates whether human epidermal growth factor receptor 2 (HER2) expression level utilizing advanced molecular positron emission tomography (PET) imaging is correlated with PTX treatment efficacy in preclinical mouse models of HER2+ breast cancer. HER2 positive (BT474, MDA-MB-361), or HER2 negative (MDA-MB-231) breast cancer cells were subcutaneously injected into athymic nude mice and PTX (15 mg/kg) was administrated. In vivo HER2 expression was quantified through [^89^Zr]-pertuzumab PET/CT imaging. PTX treatment response was quantified by [^18^F]-fluorodeoxyglucose ([^18^F]-FDG) PET/CT imaging. Spearman’s correlation, Kendall’s tau, Kolmogorov–Smirnov test, and ANOVA were used for statistical analysis. [^89^Zr]-pertuzumab mean standard uptake values (SUV_mean_) of BT474 tumors were 4.9 ± 1.5, MDA-MB-361 tumors were 1.4 ± 0.2, and MDA-MB-231 (HER2−) tumors were 1.1 ± 0.4. [^18^F]-FDG SUV_mean_ changes were negatively correlated with [^89^Zr]-pertuzumab SUV_mean_ (r = −0.5887, *p* = 0.0030). The baseline [^18^F]-FDG SUV_mean_ was negatively correlated with initial [^89^Zr]-pertuzumab SUV_mean_ (r = −0.6852, *p* = 0.0002). This study shows PTX treatment efficacy is positively correlated with HER2 expression level in human breast cancer mouse models. Molecular imaging provides a non-invasive approach to quantify biological interactions, which will help in identifying chemotherapy responders and potentially enhance clinical decision-making.

## 1. Introduction

Breast cancer is the most common cancer in women [1], with 10–15% overexpressing the human epidermal growth factor receptor 2 (HER2) [2]. The HER2 receptor (also called Neu), a member of the epidermal growth factor receptor family EGFR/ErbB, is a transmembrane glycoprotein with tyrosine kinase activity [3,4]. This receptor regulates both proliferation and differentiation of epithelial cells, and breast cancer with overexpressed HER2 tends to have increased proliferation, increased disease recurrence, and poorer survival compared to HER2− breast cancers [5]. 

Paclitaxel (PTX) is one of the most common first-line chemotherapies for the treatment of breast cancer [6,7], which induces cell apoptosis through microtubular stabilization and cell cycle arrest [8,9]. However, PTX treatment efficacy varies among patients, and it has been reported that the overall PTX response rate is between 30–60% for the treatment of metastatic breast cancer [10]. Novel discoveries of molecular mechanisms that account for ineffective response are critical in understanding breast cancer treatment. PTX is part of standard-of-care for HER2+ breast cancer patients; nevertheless, the association between HER2 and PTX treatment efficacy remains surprisingly unclear, with controversial results in both in vitro and in vivo preclinical studies [11]. In vitro studies have demonstrated that HER2 signaling is one mechanism of PTX resistance [12,13,14]; however, many clinical trials have shown HER2+ patients have a higher overall response rate than that of HER2− patients [11,15,16]. In a study with 122 metastatic breast cancer patients, the association between HER2 overexpression and clinical sensitivity to PTX was revealed, showing that 65.2% of patients with HER2+ tumors responded, compared to 35.5% of patients with HER2− tumors [15]. While most in vitro studies link HER2 expression with PTX resistance [12,13,14], a recent paper by Haghnavaz et al. shows that HER2+ cell lines exhibit lower IC50s (drug concentration that causes 50% cell death) of PTX when compared to HER2− cell lines, indicating that HER2+ cell lines show increased sensitivity to PTX when compared to HER2− cell lines [17]. In our study, we utilized controlled in vivo preclinical experiments to quantify the relationship of HER2 expression and PTX response to therapy. 

Molecular imaging allows the measurement and tracking of cellular signaling during the early course of anti-cancer treatments in a non-invasive manner. Positron emission tomography (PET) imaging is a molecular imaging modality that provides whole-body information on spatial uptake of an exogenous radiotracer (radiopharmaceutical). HER2-based PET imaging provides a global view of HER2 expression within a tumor, providing three-dimensional information on tumor heterogeneity. This offers the potential to avoid bias with tumor biopsies, which only assess information from a single point in time and a sample region of the tumor. Pertuzumab is a humanized monoclonal antibody that targets the second domain of the extracellular portion of HER2 receptors [18]. It is FDA approved to be administrated in combination with another anti-HER2 antibody, trastuzumab (Herceptin), and chemotherapy [19]. Thus, using antibody-based imaging techniques provides a potential avenue to visualize and quantify HER2 receptor overexpression. Previously, Massicano et al. reported that the non-invasive PET imaging method utilizing [^89^Zr] labeled pertuzumab has high sensitivity and specificity for HER2+ tumors and can be used to monitor response to targeted HER2 therapies [20]. Other HER2-based PET tracers include [^18^F]-aptamer [21], [^68^Ga]-NOTA-MAL-Cys-MZHER_2:342_ [22], [^64^Cu]-DOTA-ZHER_2:342_ [23], and [^89^Zr]-trastuzumab [24]. To diagnose, stage, restage, and monitor therapy responses in breast cancer, [^18^F]-fluorodeoxyglucose ([^18^F]-FDG) PET imaging is commonly used clinically as it provides a non-invasive measure of glucose metabolism within tumors [25,26,27,28]. [^18^F]-FDG PET is widely used to estimate long-term prognosis of patients in many cancer types [29], including breast cancer [30], ovarian cancer [31], lymphoma [32,33], osteosarcoma [34], and lung cancer [35]. Furthermore, it has demonstrated success in preclinical and clinical studies to predict and monitor chemotherapy efficacy [36,37,38]. These methods are clinically translatable and can provide insight into early tumor biological alterations during the course of therapy.

In this study, we examine the relationship between HER2 expression and tumor heterogeneity with PTX treatment efficacy in controlled xenografted human breast cancer models: BT474 (high HER2+), MDA-MB-361 (low HER2+), MDA-MB-231 (HER2−) tumors with molecular imaging. In vivo inter- and intra-group heterogeneity of HER2 expression was quantified with histogram analysis of standardized uptake values (SUV) of [^89^Zr]-pertuzumab PET imaging. PTX treatment responses were quantified with the measurements of tumor metabolism by using [^18^F]-FDG PET imaging. Correlations between HER2 level and PTX response were compared among and within three different tumor models. Identifying the relationship between HER2 expression and chemotherapy response provides both potential treatment targets for chemotherapy nonresponse and a clinical-relevant approach with molecular imaging to predict and guide therapeutic interventions in breast cancer.

## 2. Results

### 2.1. [^89^Zr]-Pertuzumab Measured HER2 Level In Vivo 

HER2 expression in vivo was quantified with [^89^Zr]-pertuzumab PET imaging in HER2+ and HER2− tumors prior to treatment (See Figure 1 for Experimental schema). As shown in Figure 2a, BT474 revealed a 250% higher uptake of [^89^Zr]-pertuzumab than MDA-MB-361 and a 345% higher uptake compared to MDA-MB-231 tumors. [^89^Zr]-pertuzumab mean standard uptake values (SUV_mean_) of BT474 (HER2+) tumors were 4.9 ± 1.5, MDA-MB-361 (HER2+) tumors were 1.4 ± 0.2 (*p* < 0.0001, compared with BT474), and MDA-MB-231 (HER2−) tumors were 1.1 ± 0.4 (*p* < 0.0001, compared with BT474) (Figure 2b, Appendix A showed SUV_max_ data). Histogram analysis of SUV showed BT474 tumors had broader SUV distribution than MDA-MB-361 and MDA-MB-231 (Figure 2c), indicating BT474 tumors had a more heterogenous expression of HER2. MDA-MB-361 had a right shift of SUV distribution compared to MDA-MB-231, indicating that MDA-MB-361 tumors had a higher HER2 expression than that of MDA-MB-231. Biological validation with Western blotting confirmed HER2 expression levels measured with PET imaging for the various tumor cell lines (Figure 2d). 

### 2.2. FDG PET Reveals Early Cellular Signaling of Treatment Response

SUV from [^18^F]-FDG PET imaging on days zero, three, and six were compared. Of note, tumor volumes were not significantly different among the three tumor cohorts on day zero (Figure 3a, *p* > 0.05). During the course of treatment, BT474 tumors showed a decreasing tumor size trend, while both MDA-MB-361 and MDA-MB-231 showed increasing trends; however, differences were not statistically significant (Figure 3a). Nevertheless, molecular imaging with [^18^F]-FDG-PET revealed early signaling prior to downstream changes in tumor size. [^18^F]-FDG SUV_mean_ showed significant changes from day zero and six in BT474 (*p* = 0.0376 compared to MDA-MB-231 on day six) and MDA-MB-361 tumors (*p* = 0.0036 compared to MDA-MB-231 on day six) (Figure 3b). There was a significant correlation between tumor volume and [^18^F]-FDG SUV_sum_ in BT474 (r = 0.4530, *p* = 0.0186), MDA-MB-361 (r = 0.5246, *p* = 0.0133), and MDA-MB-231 tumors (r = 0.7531, *p* < 0.0001), respectively (Appendix A). PTX treatment efficacy was further confirmed with biological validation at the experimental endpoint with Ki67 immunohistochemistry (IHC) staining (Figure 3c,d). Ki67 IHC staining showed that BT474 and MDA-MB-361 tumors had significantly reduced Ki67 positive staining compared to MDA-MB-231 tumors (*p* < 0.05) (Figure 3d). [^18^F]-FDG PET indicated earlier cellular signaling changes of treatment response to PTX compared to tumor volume measurements, and [^18^F]-FDG response was utilized to represent PTX treatment efficacy in the following analysis.

### 2.3. High HER2 Expressed BT474 Tumors Showed More Reduced FDG Uptake than Low HER2 Expressed MDA-MB-361 Tumors

HER2 expression level and PTX treatment efficacy were compared with [^89^Zr]-pertuzumab and [^18^F]-FDG PET imaging. [^89^Zr]-pertuzumab PET imaging revealed high expression of HER2 in BT474, moderate expression of HER2 in MDA-MB-361, and low expression of HER2 in MDA-MB-231, respectively (Figure 4a). From day zero to day six, [^18^F]-FDG SUV_mean_ decreased 50% in BT474 tumors, increased 50% in MDA-MB-361 tumors, and remained without significant changes in MDA-MB-231 tumors (Figure 4b,c, Appendix A shows corresponding SUV_max_ data). [^18^F]-FDG SUV_mean_ percent change from day three to day six and [^89^Zr]-pertuzumab SUV_mean_ revealed a significant, yet moderate, correlation (r = −0.5887, *p* = 0.0030, Figure 4d, Appendix A shows corresponding SUV_max_ data). By thresholding [^89^Zr]-pertuzumab at a SUV_mean_ of 2.4 (the mean value of [^89^Zr]-pertuzumab SUV_mean_), tumors were classified as high HER2 (SUV_mean_
≥ 2.4) and low HER2 (SUV_mean_ < 2.4) tumors. High HER2 tumors showed reduced [^18^F]-FDG SUV_mean_ from day zero to day six, while low HER2 tumors showed increased [^18^F]-FDG SUV_mean_ over the course of six days (Figure 4e). 

### 2.4. HER2 Level Varies within Tumor Models 

In addition to the varying levels of HER2 expression levels among the three tumor types, heterogeneous HER2 expression within tumor models was quantified with [^89^Zr]-pertuzumab PET imaging. BT474 (Figure 5a,d), MDA-MB-361 (Figure 5b,e), and MDA-MB-231 (Figure 5c,f) tumors showed inter-tumoral variability of HER2 expression levels within tumors of each xenograft model. This was demonstrated by evaluating the mean and max SUV of each tumor within each group. Among the three tumor types, BT474 showed the most variability of SUV, revealing a 344% difference between the minimum and maximum tumors, compared to the 63% and 174% difference displayed by MDA-MB-361 and MDA-MB-231, respectively (Figure 5a–c). 

### 2.5. HER2 Level Is Positively Correlated with PTX Treatment Efficacy within Tumor Models 

To explore how tumor heterogeneity of HER2 expression within a single identical tumor model affects treatment response to chemotherapy, the correlation between HER2 expression level and PTX treatment efficacy within tumor models was quantified. In BT474 tumors, low [^89^Zr]-pertuzumab SUV_mean_ tumors showed that [^18^F]-FDG SUV_mean_ remained relatively stable from day zero to day six, while high [^89^Zr]-pertuzumab SUV_mean_ tumors exhibited a dramatic decrease in [^18^F]-FDG SUV_mean_ during PTX treatment, as shown in the representative images within the group (Figure 6a,b). Histogram plot of [^18^F]-FDG SUV revealed decreased heterogeneity in high HER2 tumor (shortened SUV range in Figure 6c) but remained the same in low HER2 tumor. Low-level HER2 expressing tumors had no significant change of [^18^F]-FDG SUV distribution from day zero to day six (distance (D) = 0.2, *p* = 0.4005, Figure 6d), while high-level HER2 expressing tumors had a significant left shift of [^18^F]-FDG SUV (D = 0.375, *p* = 0.0072, Figure 6c). [^18^F]-FDG SUV_mean_ percent change from day three to day six and [^89^Zr]-pertuzumab SUV_mean_ in BT474 tumors revealed a statistically non-significant and weak correlation (r = −0.3333, *p* = 0.2595). Similar results were also found in MDA-MB-361 tumors. Low [^89^Zr]-pertuzumab uptake tumors showed increased [^18^F]-FDG SUV_mean_ from day zero to day six, while high [^89^Zr]-pertuzumab uptake tumors exhibited relatively stable in [^18^F]-FDG SUV_mean_ during PTX treatment (Appendix A). [^18^F]-FDG SUV_mean_ percent change from day zero to day six and [^89^Zr]-pertuzumab SUV_mean_ in BT474 tumors revealed a statistically non-significant and weak correlation (r = −0.4286, *p* = 0.3536, Appendix A).

### 2.6. High HER2 Tumors Had Low FDG Uptake 

The relationship between HER2 level and glucose metabolism was explored. BT474 tumors had 50% less baseline [^18^F]-FDG SUV_mean_ compared to MDA-MB-231 tumors (*p* < 0.01, Figure 7a), and MDA-MB-361 tumors had 67% of baseline [^18^F]-FDG SUV_mean_ of MDA-MB-231 tumors (*p* < 0.05, Figure 7a), indicating HER2+ tumors had low FDG uptake. Although [^89^Zr]-pertuzumab imaging and [^18^F]-FDG initial PET were conducted on different days, tumor volumes of BT474, MDA-MB-361, and MDA-MB-231 were not significantly different between imaging time points (*p* = 0.4669, 0.3698, 0.1317 for BT474, MDA-MB-361, and MDA-MB-231, respectively), therefore allowing direct comparison (Figure 3a). HER2 expression measured with [^89^Zr]-pertuzumab PET was negatively correlated with tumor glucose metabolism measured with [^18^F]-FDG (r = −0.6852, *p* = 0.0002. Figure 7b). With thresholding of [^89^Zr]-pertuzumab SUV_mean_ at 2.4, low HER2 tumors showed significantly higher [^18^F]-FDG SUV_mean_ than high HER2 tumors (*p* < 0.001. Figure 7c). With thresholding of baseline [^18^F]-FDG SUV_mean_ at 0.15, low glucose metabolism tumors showed a significant reduction in [^18^F]-FDG SUV_mean_ from day zero to day six. (*p* = 0.049. Figure 7d). Histological analysis using consecutive sections of center slices of the whole tumor further confirmed the negative correlation. IHC of GLUT1 indicated low-expressing HER2 tumors had higher GLUT1 positive staining compared to high-expressing HER2 tumors (Figure 7e,f). HER2 IHC results showed BT474 had significantly higher HER2 expression than that of MDA-MB-361 and MDA-MB-231 tumors (*p* < 0.01, Figure 7e,g). HER2 and GLUT1 IHC positive stain percentages are significantly, strongly, and negatively correlated in the three tumor models (r = −0.7617, *p* < 0.001).

## 3. Discussion

The overall goal of this study was to investigate if molecular imaging of HER2 can identify response to PTX treatment in HER2+ breast cancer. Paclitaxel is commonly used in HER2+ breast cancer patients; however, the relationship between HER2 expression and PTX efficacy has remained unclear. In this study, we proposed the potential usefulness of [^89^Zr]-pertuzumab in predicting PTX treatment efficacy. This imaging method is clinically translatable and could provide additional information to clinically utilized biopsy samples. Additionally, sequential imaging with [^18^F]-FDG provides an approach to monitor the dynamic changes of PTX response in breast cancer patients. We explored this relationship with controlled in vivo human cell line models and molecular imaging of HER2 and glucose metabolism. Positive correlations between HER2 levels and PTX efficacy with human breast cancer cell xenograft mouse models were shown, revealing that HER2 expression has a direct relationship to chemotherapy response. We measured in vivo HER2 expression with [^89^Zr]-pertuzumab and examined its inter- and intra- tumoral heterogeneity, revealing that there is variability within each model. Additionally, [^18^F]-FDG revealed early cellular signal changes prior to tumor volume changes in response to PTX treatment efficacy, which was confirmed with biological validation with IHC staining of Ki67. Our results indicated high HER2 expressing tumors had a more significant reduction in tumor metabolism after PTX treatment. This phenomenon was observed among the three human cell xenograft tumor models and within individual tumor models. This study suggested high HER2 expressing tumors will have a better therapeutic response to PTX treatment compared to low HER2 expressing tumors, thus providing a potential indicator that could help in clinical decision-making in the future.

Clinically, there are two commonly used methods to detect HER2 levels: (1) immunohistochemistry, which detects HER2 protein, and (2) fluorescence in situ hybridization, which detects the copy number of the HER2 gene. Nevertheless, both methods rely on invasive biopsies, which may not represent overall tumor status due to intratumoral heterogeneity, providing a single snapshot in time. Molecular PET imaging of HER2 has been developed with radiopharmaceuticals which includes small molecules (e.g., peptides), antibodies, and affibodies [39]. HER2 PET imaging can provide spatial and temporal variations in tumor changes and has been shown to reveal response to anti-HER2 therapies [20]. In this study, we quantified both the inter- and intra-group heterogeneity of HER2 expression variations in preclinical models of HER2+ breast cancer with molecular imaging. [^89^Zr]-pertuzumab PET imaging revealed BT474 tumors had much higher HER2 expression levels than MDA-MB-361 and MDA-MB-231 (Figure 2). Further, BT474 tumors showed large variability in HER2 expression levels extracted from antibody-based PET imaging, among which the high HER2 tumors can have as much as three times higher HER2 expression compared to low expression HER2 tumors (Figure 5a).

There has been a controversy on whether HER2 expression is associated with PTX resistance in the literature. Yu et al. [13,40] reported high HER2 expressing cell lines had high PTX IC50; however, Haghnavaz et al. [17] showed results where high HER2 expressing cell lines had low PTX IC50. In clinical trials, some studies showed HER2+ tumors had a better overall response rate to PTX compared to HER2− tumors [15,41]. However, other clinical trials revealed there was no association between HER2 and PTX response [42]. In our study, we demonstrated that high HER2 expressing tumors (BT474) had better responses to PTX compared to low HER2 expressing MDA-MB-361 tumors (Figure 4). Also, within just the BT474 (Figure 6) or MDA-MB-361 (Appendix A) tumor cohort, high HER2 expression showed a moderate but non-significant correlation with better PTX response. The non-significance could be due to the small sample size. Statistical power analysis with G*power software showed that under the assumption of a correlation of −0.3333 or −0.4286 between HER2 and FDG changes within BT474 or MDA-MB-361 tumors, a sample size of 68 (BT474) or 40 (MDA-MB-361) would be required in order to have 80% power to detect this effect at α = 0.05. Overall, our results support that HER2 expression is positively correlated with PTX efficacy. 

Currently, there is no difference in prescribed treatment regimens between clinically high and low HER2+ breast cancer patients. In our study, we found high HER2 expression tumors, BT474, responded more efficiently to paclitaxel, with reduced glucose metabolism over the course of a week as demonstrated with decreased FDG SUV_mean_ (Figure 3b and Figure 4a) and FDG SUV_max_ (Appendix A) during treatment. Conversely, HER2+ tumors with low HER2 levels had increased FDG SUV_mean_ (Figure 3b and Figure 4a) and FDG SUV_max_ (Appendix A) after paclitaxel treatment. These results provided a preliminary assessment that paclitaxel might be more effective in high HER2 expressing HER2+ tumors but not as effective in tumors with low HER2 expression. The potential thresholding quantified from this study of [^89^Zr]-pertuzumab to separate out this response was determined as a SUV_mean_ of 2.4 (Figure 4e). 

A negative correlation was identified between HER2 level and FDG uptake (Figure 7), and the HER2− (MDA-MB-231) and low HER2 (MDA-MB-361) tumors had higher GLUT1 expression than high HER2 tumors (BT474) (Figure 7e). This revealed preliminary evidence that high HER2 expression tumors had low glucose uptake capacity and enhanced paclitaxel anti-tumoral activity. FDG is internalized by glucose transporters (Glut 1–13), phosphorylated by HK to FDG-6-phosphate, and then trapped in the cells [43]. GLUT1 expression has been shown to be correlated with tumor FDG uptake in salivary gland pleomorphic adenomas [44], esophageal squamous cell carcinoma [45], and high-risk prostate cancer [46]. Choi et al. found the overexpression of GLUT1 was associated with triple negative breast cancer and not HER2 breast cancer patients [47]. Deng et al. also found GLUT1 overexpression was associated with high histological grade ER−, PR− breast cancer patients; however, no significant correlation was seen between GLUT1 level and HER2 status [48]. Nevertheless, several studies showed HER2 signaling enhances metabolic enzymes and promoted glucose uptake [49,50,51]. O’Neal et al. showed in vitro MCF10A-HER2 cells had increased glycolysis compared to MCF10A [52]. It is hypothesized that tumors with higher HER2 have lower GLUT1 (and therefore lower FDG uptake) and that the underlying glycolysis in the tumors could affect sensitivity to paclitaxel treatment. Paclitaxel-resistant cells were shown to have high glycolysis rates due to a unique Warburg-like metabolism [53,54]. Surov et al. [55] reported the association between FDG uptake and Ki67 histological staining with a meta-analysis of breast cancer patients. However, in our study, no significant association between FDG uptake and Ki67 staining was observed, which could be due to differences in treatment regimen or the analysis of histology data. Our quantitative assessment of the histology data used whole tumor cross-sectional analysis, which allowed for the entire tumor heterogeneity to be included in Ki67 quantification, whereas traditional methods for histological analysis utilize biopsy samples or hot spot analysis.

Limitations of the study include the lack of long-term tumor response to paclitaxel; however, imaging and immunohistochemistry biological validation revealed early cellular response to treatment prior to downstream changes in tumor size. Longitudinal response kinetics to paclitaxel for these tumor models have been previously reported for BT474 [56], MDA-MB-361 [57], and MDA-MB-231 [58] in vivo, revealing that these tumors do respond to paclitaxel with eventual decreases in tumor volume. Our study evaluated the molecular changes in the tumor, which occurred prior to downstream alterations in tumor size. While the tumor size did not show significant changes during one-week treatment, FDG uptake and Ki67 immunohistochemistry staining showed significant biological alterations (Figure 3). Furthermore, preliminary results revealing the correlation between HER2 expression and glucose uptake warrant further investigation. Exploration on the underlying mechanism of how HER2 expression affects glucose transporters and glycolysis may provide further information on the heterogeneity of chemotherapy response. Finally, an additional limitation to the study is that [^18^F]-FDG PET (day zero) and [^89^Zr] pertuzumab PET (day −8) were compared on different days (Figure 7). Although there were no statistical changes in tumor volumes during that timeframe (Figure 3a), we acknowledge there may be some biological alterations in the tumors during this eight-day time window. The timeframe between [^18^F]-FDG PET and [^89^Zr] pertuzumab PET was a necessity as the half-life of Zr-89 is 3.3 days, and pertuzumab, as an antibody, requires time to distribute in the body prior to imaging. Therefore, for effective imaging and wash-out, a delay between imaging is required. Biological validation at the experimental endpoint allowed for further validation of GLUT1 and HER2 IHC staining of consecutive sections of the center slice of the whole tumor (Figure 7h).

Overall, this study showed HER2 expression measured by [^89^Zr]-pertuzumab PET is positively correlated with paclitaxel treatment efficacy measured by FDG PET. Intergroup and intragroup heterogeneity of HER2 expression in tumors can be extracted with [^89^Zr]-pertuzumab PET and utilized to help assess chemotherapy response. High HER2 and low HER2 tumors show variation in response to paclitaxel, which may potentially help in guiding the clinical treatment regimen of HER2+ breast cancer patients. Finally, the imaging methods used in the paper are clinically translatable and have the potential to be applied prior to treatments in patients and to help predict the eventual therapeutic response.

## 4. Materials and Methods 

### 4.1. Cell Culture 

HER2+ breast cancer cells BT474 and MDA-MB-361 and triple-negative breast cancer cells (MDA-MB-231) were purchased from American Type Culture Collection (ATCC). BT474 cells were cultured in Dulbecco’s Modified Eagle Medium (DMEM) (Fisher, 31-053-036) with 10% Fetal Bovine Serum (FBS) (R&D, S12450H), 29.20 mg/mL L-glutamine (Corning, 25-005-CI), 1mM sodium pyruvate (Gibco, 11360-070), and 0.01 mg/mL insulin (Gibco, 12585-014). MDA-MB-361 cells were cultured in DMEM with 20% FBS, 29.20 mg/mL L-glutamine, and 1mM sodium pyruvate. MDA-MB-231 cells were cultured in DMEM with 1% FBS, 29.20 mg/mL L-glutamine, and 1mM sodium pyruvate. All cells were maintained in an incubator at 37 °C with 5% CO_2_ and cultured to 80–90% confluence.

### 4.2. Mouse Xenograft and Tumor Volume Measurement 

Animal experiments were reviewed and approved by the Institutional Animal Care and Use Committee (IACUC) at the University of Alabama at Birmingham. Cell counts of 1 × 10^7^ BT474 (*n* = 9), 6 × 10^6^ MDA-MD-361 (*n* = 7), or 2 × 10^6^ MDA-MD-231 (*n* = 7) were injected serum-free DMEM media with 20% growth factor-reduced Matrigel (Corning, 356232) in the flank of nude athymic mice at 6 weeks of age. Both MDA-MB-361 and BT474 models are well-established models of HER2+ breast cancer with known HER2 level differences [59,60,61,62]. Tumor sizes were measured with a caliper weekly until reaching an average of 200 mm^3^, and then they were enrolled into the study. MDA-MB-231, MDA-MB-361, and BT474 tumors were 423 ± 155 mm^3^, 382 ± 118 mm^3^, 299 ± 127 mm^3^. There was no significant difference at the initial time point between these three models (one-way ANOVA: *p* > 0.05). Following the initiation of the study, tumors were measured every 3 days. Tumor volumes were calculated as $V=\frac{1}{6}\pi {\left(transverse\;diameter\right)}^{2}\left(longitudinal\;diameter\right)$. In vivo HER2 expression levels were determined by [^89^Zr]-Pertuzumab PET/CT imaging at one week prior to initiation of treatment and confirmed with Western blot analysis. PTX (15 mg/kg) was administered via i.v. on days 0 and 3. In vivo tumor metabolism was quantified by [^18^F]-FDG PET/CT imaging on days 0, 3, and 6.

### 4.3. Experimental Schema

The experimental timeline is shown in Figure 1. Briefly, BT474 (1 × 10^7^), MDA-MB-361 (6 × 10^6^), or MDA-MB-231 (2 × 10^6^) human breast cancer cells were subcutaneously injected into athymic nude mice. Mice were enrolled in the experiments when tumors exceeded 200 mm^3^. Approximately 100 μCi [^89^Zr]-pertuzumab (4 µCi/µg) were injected i.v. into mouse. PET/CT imaging was acquired on day −8. [^18^F]-FDG PET/CT imaging was acquired on day 0 (baseline), day 3, and day 6. PTX (15mg/kg) was administrated via i.v on day 0 and 3. On day 6, tumor samples were collected for histology analysis.

### 4.4. PET Imaging 

#### 4.4.1. [^89^Zr]-pertuzumab PET/CT Imaging

Three micrograms of pertuzumab (Roche, NDC 50242-145-01) were conjugated with 0.4 mg chelator deferoxamine (DFO) at 10 mg/mL. [^89^Zr] was supplied onsite by the University of Alabama at Birmingham cyclotron facility. DFO-pertuzumab was labeled with [^89^Zr] at 4 μCi/μg. 100 μCi per mouse [^89^Zr]-pertuzumab were i.v. injected seven days before imaging. PET/CT imaging was performed with a GNEXT small animal PET/CT (Sofie, Dulles, VA, USA) for a 30 min PET static scan (6 days post-injection). CT imaging was acquired with 80 kvp and 120 μA Bin2 parameters. 

#### 4.4.2. [^18^F]-fluorodeoxyglucose (FDG) PET/CT Imaging 

Mice were fasted overnight prior to imaging. 100 μCi per mouse [^18^F]-FDG was injected 1 h prior to a 10 min static PET scan. CT imaging was acquired with 80 kvp and 120 μA Bin2 parameters. 

#### 4.4.3. Image Analysis

PET and CT images were registered, and the whole tumor region of interest (ROI) was drawn based on the anatomical CT images. Mean, standard deviation, sum, and frequency histogram of standard uptake value (SUV) were quantified (VivoQuant software, Boston, MA, USA) with the equation $SUV=\frac{C}{dose/weight}$ where C, is the tissue radioactivity concentration; dose is the injected dose; weight is mouse body weight. 

#### 4.4.4. Western Blot Evaluation of HER2 Expression 

BT474, MDA-MB-361, and MDA-MB-231 breast cancer cells were washed and lysed on ice with RIPA buffer supplemented with protease inhibitor (Roche Applied Science, Indianapolis, IN). Lysates were centrifuged at maximum speed at 4 °C and collected for protein quantification assay (Pierce^TM^ BCA protein assay kit, 23225, Themo Fisher Scientific) via the Nanodrop 2000c spectrophotometer (Thermo Fisher Scientific, Waltham, MA, USA). For each model, 20 μg of protein was prepared in a solution of sodium dodecyl sulfate and β-mercaptoethanol and boiled for 5 min at 95 °C. Samples were run on a NuPAGE Bis-Tris gel and transferred to a PVDF membrane. The membrane was blocked in 10% dry milk diluted in TBST, probed with HRP conjugated mouse anti-human β-actin overnight at 4 °C and developed with the Amersham ECL Western blot detection system (GE healthcare, Buckinghamshire, UK). Membranes were developed and visualized with an SRX-101A Medical Film Processor (Konica Minolta Medical and Graphic, Inc., Shanghai, China). The membrane was stripped, washed with tris-buffered saline with 0.01% tween (TBST) buffer, and re-probed with Rabbit anti-human HER2/ErbB2 primary antibody (2242S, Cell Signaling Technology, Danvers, MA, USA) overnight. The membrane was then washed, incubated with HRP conjugated goat anti-rabbit IgG secondary antibody (7074S, Cell Signaling Technology, Danvers, MA, USA) for 1 h. After a final wash, the membrane was redeveloped and visualized for HER2 expression. Quantification of bands was conducted with the Image Studio Lite program, version 5.0 (LI-COR Biosciences, Lincoln, NE, USA). Western blot quantification was done with ImageJ (Fiji contributors).

#### 4.4.5. Immunohistochemistry Staining 

Tumor samples were fixed with 10% formalin for 24 h at room temperature, washed with PBS for 30 min, gradient dehydrated with 30% ETOH, 50% ETOH, and 70% ETOH for half an hour in each solution. Paraffin embedding and 5 μm sections of the whole tumor center slices were performed in the UAB histology core facility. H&E staining was conducted as previously reported [63]. Paraffin sections were dewaxed with xylene 10 min twice, then gradient rehydrated with 100% ETOH × 2, 95% ETOH, 70% ETOH, 50% ETOH, and dH_2_O × 2 for 2 min each. Antigen Retravel was conducted with citrate acid and steamed in a rice cooker for 30 min. Then, the slides were emerged with 3% H_2_O_2_ for 10 min and blocked with 1% BSA in PBS with 0.05% Triton 100 for 30 min at room temperature. Primary antibodies were incubated overnight at 4 °C: anti-Ki67 (abcam, ab16667, 1:200), anti-Her2 (Cell Signaling Technology, 2242, 1:200), and anti-GLUT1 (abcam, ab115730, 1:200). Anti-Rabbit-IgG (GTX83399, GeneTex) was used as a secondary antibody and incubated at room temperature for 30 min. DAB substrate kit (Vector, SK-4105) was used to develop the staining. Images were taking with EVOS M7000 Imaging System under simple mode with brightness at 0.005 and a 20× supersonic wave drive lens. 

#### 4.4.6. Immunohistochemistry Analysis 

Analysis of positive IHC staining of the whole section was completed through automated custom MATLAB algorithms. Auto-segmentation of each image was initially conducted in order to isolate the tumor area from background white space using optimal thresholding, and the total number of pixels within the tumor was quantified. After using k-means to cluster the tumor image into 6 segments, a viable positively stained tissue mask was rendered. Color thresholding was conducted on the resulting color masks containing positively stained pixels to remove noise variation and pick out true positive staining. Then, these images were binarized for the summation of positively stained tissue pixels. The percentage of positive staining was defined as the total number of positively stained pixels divided by the total number of tumor pixels with noise and background removed and converted to a percentage.

#### 4.4.7. Statistical Analysis

Kolmogorov–Smirnov, Spearman’s correlation, Kendall’s tau correlation, and ANOVA were used as statistical analysis. Specifically, HER2 expression levels and histological analysis were tested with one-way ANOVA and Tukey’s post hoc test. Tumor volumes and FDG SUV_mean_ from day 0 to day 6 were tested with mixed ANOVA and Tukey’s post hoc test. Correlation tests with three mouse models were Spearman’s correlation. Correlation tests within mouse models were done with Kendall’s tau correlation due to the small sample size. Histogram plots were tested with Kolmogorov–Smirnov test. Figure 4e and Figure 7c,d were tested with an unpaired *t*-test. A *p*-value of <0.05 was considered statistically significant. Non-significant, *p* > 0.05; * *p* < 0.05; ** *p* < 0.01; *** *p* < 0.001.

## Figures and Tables

**Figure 1 molecules-26-01568-f001:**
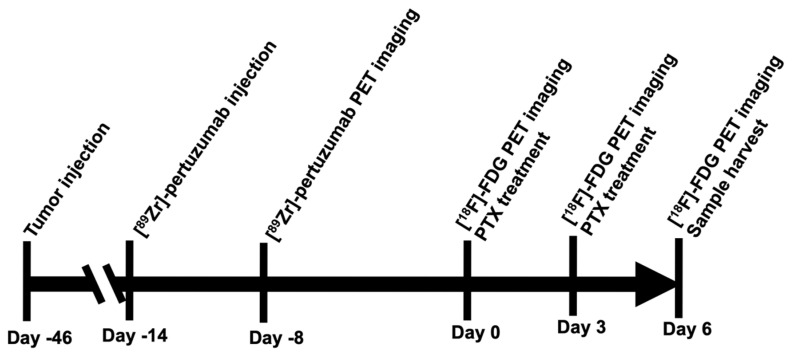
Experimental scheme. Schematic diagram of experimental timeline.

**Figure 2 molecules-26-01568-f002:**
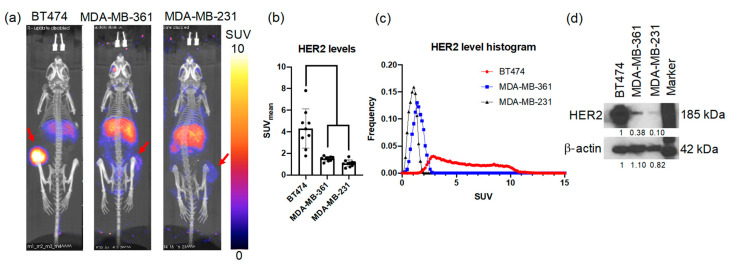
[^89^Zr]-pertuzumab PET imaging indicates HER2 levels in three human breast cancer xenograft mouse models. (**a**) Representative [^89^Zr]-pertuzumab PET images in maximum intensity projection (MIP) view. Red arrows point to tumors. (**b**) Mean standard uptake values (SUV_mean_) of [^89^Zr]-pertuzumab PET. ANOVA and Tukey’s post hoc test: **** *p* < 0.0001. (**c**) Histogram plot of SUV of the three mouse models indicating the heterogeneity of HER2 expression as measured by PET imaging. (**d**) Western blotting of HER2 expression levels in BT474, MDA-MB-361, and MDA-MB-231 cells.

**Figure 3 molecules-26-01568-f003:**
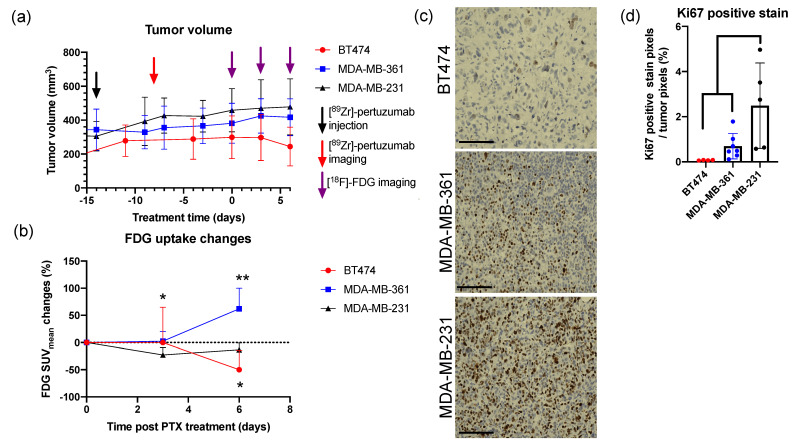
[^18^F]-fluorodeoxyglucose ([^18^F]-FDG) PET imaging reveals early cellular signal of treatment response. (**a**) Caliper measurements are shown for tumor volumes of BT474, MDA-MB-361, and MDA-MB-231. Mixed ANOVA and Dunnett’s multiple comparisons test: non-significant. (**b**) The changes of the mean of standard uptake value (SUV_mean_) of [^18^F]-FDG PET in BT474, MDA-MB-361, and MDA-MB-231 tumors. Mixed ANOVA and Tukey’s post hoc test: * *p* < 0.05; ** *p* < 0.01. (**c**) Representative Ki67 immunohistochemistry (IHC) staining in three tumor types from the center slice of the whole tumor section. Scale bar: 125 μm. (**d**) Quantification of IHC of the center slice of the whole tumor section revealed that BT474 and MDA-MB-361 tumors had significantly reduced Ki67 positive staining compared to MDA-MB-231 tumors. One-way ANOVA and Tukey’s post hoc test: * *p* < 0.05.

**Figure 4 molecules-26-01568-f004:**
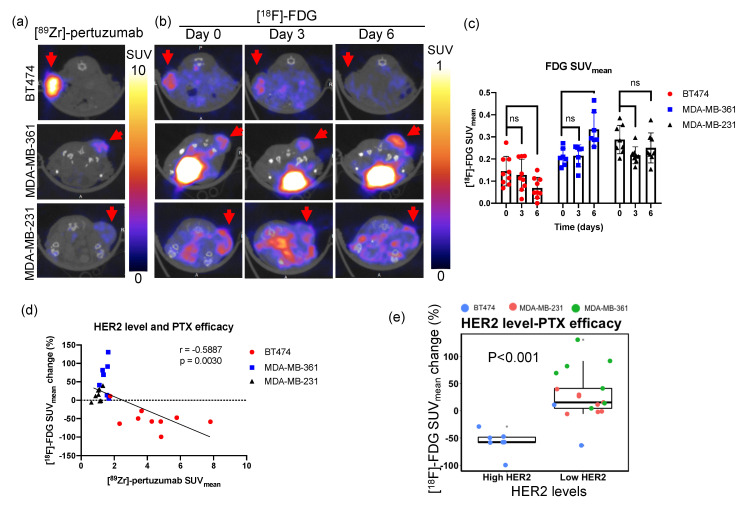
HER2 expression level is correlated with paclitaxel treatment efficacy in three human breast cancer xenograft mouse models. (**a**) Representative [^89^Zr]-pertuzumab PET images of BT474, MDA-MB-361, and MDA-MB-231 tumors are shown as a transverse cross-section through the mouse. Red arrows denote tumors. (**b**) Representative [^18^F]-FDG PET images of BT474, MDA-MB-361, and MDA-MB-231 tumors from day zero to day six in transverse section. Red arrows denote tumors. (**c**) SUV_mean_ of [^18^F]-FDG in BT474, MDA-MB-361, and MDA-MB-231 tumors from day zero to day six. Mixed ANOVA and Tukey’s post hoc test: ns, non-significant; * *p* < 0.05. (**d**) The percent change of SUV_mean_ of [^18^F]-FDG from day three to day six is negatively correlated with SUV_mean_ of [^89^Zr]-pertuzumab. Spearman’s correlation: r = −0.5887, *p* = 0.0030. (**e**) High HER2 (SUV_mean_
≥ 2.4) tumors show a more reduced SUV_mean_ of [^18^F]-FDG from day zero to day six than low HER2 (SUV_mean_ < 2.4) tumors. Unpaired *t*-test: *p* < 0.001.

**Figure 5 molecules-26-01568-f005:**
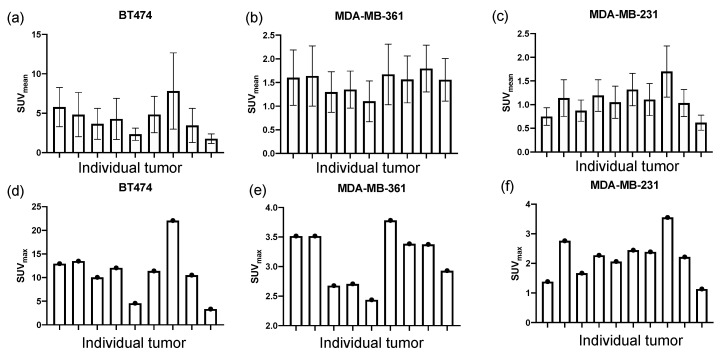
[^89^Zr]-pertuzumab PET imaging reveals HER2 expression varies within human breast cancer xenograft mouse models. (**a**–**c**) HER2 level measured by SUV_mean_ of [^89^Zr]-pertuzumab varies within BT474 (**a**), MDA-MB-361 (**b**), and MDA-MB-231(**c**) tumors. (**d**–**f**) Bar charts with individual values of SUV_max_. HER2 level measured by SUV_max_ of [^89^Zr]-pertuzumab varies within BT474 (**d**), MDA-MB-361 (**e**), and MDA-MB-231(**f**) tumors.

**Figure 6 molecules-26-01568-f006:**
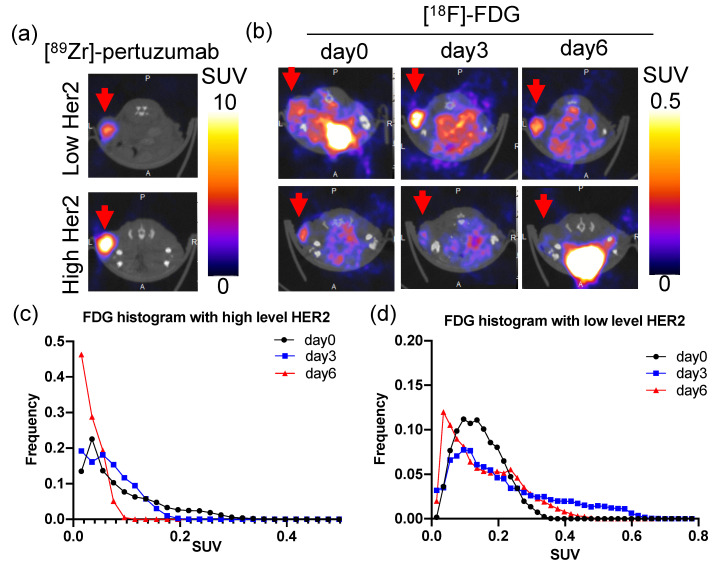
HER2 expression level is correlated with paclitaxel treatment efficacy in BT474 tumor model. (**a**) Representative [^89^Zr]-pertuzumab PET images of BT474 tumors with high and low HER2 levels in transverse section. Red arrows point to tumors. (**b**) Representative [^18^F]-FDG PET images of BT474 tumors with high and low HER2 expression from day zero to day six in transverse orientation. Red arrows point at tumors. (**c**) Histogram plot of [^18^F]-FDG of BT474 tumor with high HER2 expression from day zero to day six. Kolmogorov–Smirnov test: day zero vs day three: *p* = 0.1641; day zero vs day six: *p* = 0.0072; day three vs day six: *p* = 0.9135. (**d**) Histogram plot of [^18^F]-FDG of BT474 tumor with low HER2 expression from day zero to day six. Kolmogorov–Smirnov test: day zero vs day three: *p* = 0.0015; day zero vs day six: *p* = 0.4005; day three vs day six: *p* = 0.0971.

**Figure 7 molecules-26-01568-f007:**
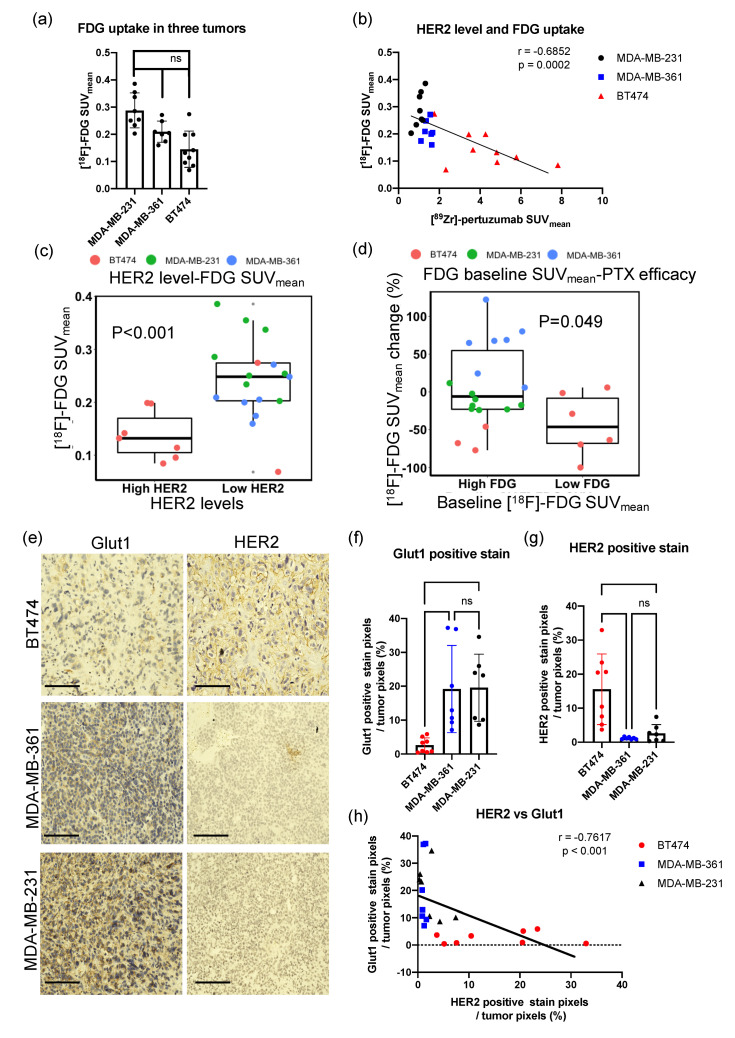
High HER2 expression tumors have low glucose metabolism and are more sensitive to PTX treatment. (**a**) Baseline [^18^F]-FDG SUV_mean_ of three tumor types. ANOVA and Tukey’s post hoc test: * *p* < 0.05; ** *p* < 0.01. (**b**) The baseline of SUV_mean_ of [^18^F]-FDG is negatively correlated with SUV_mean_ of [^89^Zr]-pertuzumab in three tumor models. Spearman’s correlation: r = −0.6852, *p* = 0.0002. (**c**) High HER2 (SUV_mean_
≥ 2.4) tumors had lower baseline SUV_mean_ of [^18^F]-FDG than low HER2 (SUV_mean_ < 2.4) tumors. Unpaired *t*-test: *p* < 0.001. (**d**) Low glucose metabolism (SUV_mean_ < 0.15) tumors had more reduced SUV_mean_ of [^18^F]-FDG from day zero to day six compared to high glucose metabolism (SUV_mean_
≥ 0.15) tumors. Unpaired *t*-test: *p* = 0.049. (**e**) Representative images of IHC of GLUT1 and HER2 from the center slice of the whole tumor consecutive sections. Scale bar: 125 μm. (**f**,**g**) Quantification of Ki67 (**f**) and GLUT1 (**g**) IHC staining of the whole tumor section shows MDA-MB-231 and MDA-MB-361 had significantly higher GLUT1 expression and lower HER2 expression than that of BT474 tumors (*p* < 0.05). One-way ANOVA and Tukey’s post hoc test: ns, non-significant; * *p* < 0.05; ** *p* < 0.01. (**h**) HER2 and GLUT1 IHC positive stain percentages were negatively correlated in the three tumor models. Spearman’s correlation: r = −0.7617, *p* < 0.0001.

## Data Availability

Please contact the corresponding author for reasonable requests for data that was generated during the described experiments.

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
