# Peer review of "[89Zr]-Pertuzumab PET Imaging Reveals Paclitaxel Treatment Efficacy Is Positively Correlated with HER2 Expression in Human Breast Cancer Xenograft Mouse Models"

_molecules, 2021, doi:10.3390/molecules26061568_

Round 1
Reviewer 1 Report
The paper is well written, focused on a very interesting topic. Figures are illustrative.
I suggest to accept the paper in its current form.
Reviewer 2 Report
Monoclonal Antibody, paclitaxel is commonly used in patients with HER2+ breceptor expression. However, up to now relationship between HER2 expression and paclitaxel efficacy has not been studied. In the manuscript authors proposed application of PET receptor imaging with [89Zr]-pertuzumab for paclitaxel treatment efficacy. Results of the studies indicated that HER2 expression measured by [89Zr]-pertuzumab PET is well correlated with paclitaxel treatment efficacy. This is very important information as it will avoid invasive methods of testing HER2 expression such as biopsy. Information is also obtained about heterogeneity of HER2 receptors on the tumor. The publication is well written and I have no significant comments on the methodology and form of the manuscript. The publication is well written and I have no significant comments on the methodology and form of the manuscript. However, I propose to conduct cellular studies on receptor affinity of pertuzumab and cytotoxicity of paclitaxel on the cells that were tested in-vivo. Also, cytotoxicity studies on spheroids can be very useful. However, I agree to acctept of the publication despite the lack of cellular studies.
This manuscript is a resubmission of an earlier submission. The following is a list of the peer review reports and author responses from that submission.
Round 1
Reviewer 1 Report
In this study, authors investigated the relation between HER2 expression of tumors and the response for paclitaxel treatment in xenograft models using Zr-89 pertuzumab and F-18 FDG PET images. After careful reading of this manuscript, I found that the protocol for imaging and treatment, and results are not very convincing. I have a few comments for this study below.
Major comments>
- Paclitaxel treatment and F-18 FDG PET imaging were started 2 weeks after Zr-89 pertuzumab injection. Significant changes in tumor status may happen in this period. It is not reasonable to compare two images obtained at fortnightly interval (Fig 7). Comparison between pretreatment HER2 expression and post-treatment GLU-1 expression is not reasonable either.
- Tumor size didn’t change for all 3 groups by 1 week-paclitaxel treatment. Why didn’t authors treat tumors longer? I don’t agree that need for histologic validation can be a reason for this short-term treatment. Is there any reference for 1 week treatment for paclitaxel? Why did authors administer paclitaxel on the day of tumor removal, which effect might not occur?
- How big were tumor sizes of 3 groups on the day of starting treatment (in actual volume, not %)? Is there any difference in tumor size among 3 groups? (In fig 4, FDG uptake on D0 was different among 3 groups). Tumor size at the start of treatment can also have influence on the therapeutic response.
- (In fig 3) FDG uptake (MDA-MB-361 > MDA-MB-231 > BT474) and Ki-67 (MDA-MB-231 > MDA-MB-361 > BT474) showed no correlation. According to the meta-analysis for FDG-PET in breast cancer (Trans Oncol 12:375-380), FDG uptake and Ki-67 correlates moderately. Please discuss on this.
- On the correlation between HER2 expression and paclitaxel response, authors mentioned that their results provide mechanistic information (”Introduction”). What do you think is the possible mechanism for this positive correlation?
Minor comments>
- (line 73 and 76) Manufacturers of drugs are shown in “Introduction”. It is recommended that they are shown in “Materials and Methods”.
- (line 90-101) In “Introduction”, the purpose of the study is too lengthy. All major experiments were listed. Please specify the most important purpose of this study.
- In fig 5, SUVmax and SUVmean of Zr-89 pertuzumab for individual tumors show mean and standard deviation. Did authors draw ROIs on whole volumes of tumors? Please explain in detail.
- Figure 5 (a)~(c) and (d)~(f) have different types of error bars. Is there any reason for this difference?
- Authors designed a MDA-MB-361 model as a weak positive control. It is unnecessary to divide BT474 model into low HER2 and high HER2 subgroups. If you want to use subgroups of BT474, MDA-MB-361 model is not necessary. In addition, figure 4 and 6 are somewhat overlapped. Please arrange figures not to show duplicated data.
Author Response
Dear Editor and Reviewer,
Thank you for the opportunity to submit a revised manuscript and for the time you have committed to improving the study. We appreciate your efforts and believe we have addressed all of your points as indicated below and in the revised manuscript.
Reviewer #1:
In this study, authors investigated the relation between HER2 expression of tumors and the response for paclitaxel treatment in xenograft models using Zr-89 pertuzumab and F-18 FDG PET images. After careful reading of this manuscript, I found that the protocol for imaging and treatment, and results are not very convincing. I have a few comments for this study below.
Response: Thank you for your comments. We appreciate your critical review and have addressed your comments point by point, as shown below.
Major Comments:
Paclitaxel treatment and F-18 FDG PET imaging were started 2 weeks after Zr-89 pertuzumab injection. Significant changes in tumor status may happen in this period. It is not reasonable to compare two images obtained at fortnightly interval (Fig 7). Comparison between pretreatment HER2 expression and post-treatment GLU-1 expression is not reasonable either.
Response: Thank you for your comment. We agree that the tumor may change between Zr-89 pertuzumab injection and F-18 FDG PET imaging, as the tumor is continuing to grow during this time; however, this protocol is clinical translatable, as breast cancer patients receive biopsies of their tumor prior to treatment planning (which can be on the order of weeks to months). Additionally, the design of this protocol is necessary as the half-life of Zr-89 is 3.3 days, and pertuzumab, as an antibody, requires time to distribute in the body, therefore for effective imaging and wash-out, a two-week requirement is needed prior to F-18 FDG PET imaging after Zr-89 pertuzumab injection. To strengthen the imaging data that was acquired, we included new HER2 immunohistochemistry data, which shows a significantly strong correlation between HER2 and GLUT1 (r=-0.7617, p<0.001, Figure 7h). Additionally, we have added this time difference as a limitation to the study in our discussion “Finally, an additional limitation to the study is that [18F]-FDG PET at day 0 and [89Zr] pertuzumab PET at day -8 were compared (Figure 7), and there may be biological changes in the tumor during this eight-day time window. This, however, was a necessity as the half-life of Zr-89 is 3.3 days, and pertuzumab, as an antibody, requires time to distribute in the body. Therefore, for effective imaging and wash-out, a delay between imaging is required. GlUT1 and HER2 IHC staining of consecutive sections of the center slice of the whole tumor were compared at the conclusion of the study to provide additional information (Figure 7h).”.
- Tumor size didn’t change for all 3 groups by 1 week-paclitaxel treatment. Why didn’t authors treat tumors longer? I don’t agree that need for histologic validation can be a reason for this short-term treatment. Is there any reference for 1 week treatment for paclitaxel? Why did authors administer paclitaxel on the day of tumor removal, which effect might not occur?
Response: Thank you for your comment. Longitudinal response kinetics to paclitaxel for these tumor models have been previously reported for BT474 [1], MDA-MB-361 [2], and MDA-MB-231 [3] tumor growth in vivo, revealing that these tumors do respond to paclitaxel with eventual decreases in tumor volume. Our study evaluated the molecular changes in the tumor, which occurs prior to downstream alterations in tumor size. While the tumor size did not show significant changes during one-week treatment, FDG uptake and Ki67 immunohistochemistry staining showed significant biological alterations. We have modified the “Discussion” (line 333-341) to expand upon this and clarify any confusion. Short-term treatment and imaging experiments were designed to predict eventual treatment efficacy, which would be both clinically translatable and potentially helpful in guiding clinical regimens for individual patients in the future. We apologize for the typo in the experimental scheme description in “Methods” and we have corrected it in the text (line 390-391). We respectfully note that we administrated paclitaxel only on days 0 and 3 and not the removal day (Figure 1 and Methods line 382-383 and line 390-391).
- How big were tumor sizes of 3 groups on the day of starting treatment (in actual volume, not %)? Is there any difference in tumor size among 3 groups? (In fig 4, FDG uptake on D0 was different among 3 groups). Tumor size at the start of treatment can also have influence on the therapeutic response.
Response: Thank you for your comment. We have added the initial tumor sizes in “Methods” (line 377-379): “MDA-MB-231, MDA-MB-361, and BT474 tumors were 423 ± 155 mm3, 382 ± 118 mm3, 299 ± 127 mm3. There was no significant difference in tumor volumes at the initial time point between these three models (one-way ANOVA: p>0.05).” In Figure 4, the SUVmean data is normalized to the tumor size.
- (In fig 3) FDG uptake (MDA-MB-361 > MDA-MB-231 > BT474) and Ki-67 (MDA-MB-231 > MDA-MB-361 > BT474) showed no correlation. According to the meta-analysis for FDG-PET in breast cancer (Trans Oncol 12:375-380), FDG uptake and Ki-67 correlates moderately. Please discuss on this.
Response: Thank you for your suggestion. We have added this point to our “Discussion” (line 326-332): “Surov et al. [4] reported the association between FDG uptake and Ki67 histological staining with a meta-analysis of breast cancer patients. However, in our study, no significant association between FDG uptake and Ki67 staining was observed, which could be due to differences in treatment regimen or the analysis of histology data. Our quantitative assessment of the histology data was whole tumor cross-sectional analysis which allows for the entire tumor heterogeneity to be included in Ki67 quantification, whereas traditional methods for histological analysis utilize biopsy samples or hot spot analysis.”
- On the correlation between HER2 expression and paclitaxel response, authors mentioned that their results provide mechanistic information (”Introduction”). What do you think is the possible mechanism for this positive correlation?
Response: Thank you for the comment. It is hypothesized that tumors with higher HER2 have lower GLUT1 (and therefore lower FDG uptake) and that the underlying glycolysis in the tumors could affect sensitivity to paclitaxel treatment. We have expanded the “Discussion” to include this (line 322-326).
Minor Comments:
- (line 73 and 76) Manufacturers of drugs are shown in “Introduction”. It is recommended that they are shown in “Materials and Methods”.
Response: Thank you, we have amended the text as suggested.
- (line 90-101) In “Introduction”, the purpose of the study is too lengthy. All major experiments were listed. Please specify the most important purpose of this study.
Response: We have shortened the purpose of the study paragraph as suggested.
- In fig 5, SUVmax and SUVmean of Zr-89 pertuzumab for individual tumors show mean and standard deviation. Did authors draw ROIs on whole volumes of tumors? Please explain in detail.
Response: Thank you for the comment. The ROIs are drawn on the whole volume of tumors according to CT images. The related method section (line 405) has been updated. We apologize for the confusion in figure 5. Standard deviation was calculated with SUVmean only.
- Figure 5 (a)~(c) and (d)~(f) have different types of error bars. Is there any reason for this difference?
Response: We apologize for the confusion in figure 5. (a)-(c) have error bars; however, (d)-(f) are bar charts with individual value. This is required as the SUVmax per tumor is a single value, whereas the SUVmean is taking the average of the entire tumor, and a standard deviation can be calculated. We have clarified this in the figure caption.
- Authors designed a MDA-MB-361 model as a weak positive control. It is unnecessary to divide BT474 model into low HER2 and high HER2 subgroups. If you want to use subgroups of BT474, MDA-MB-361 model is not necessary. In addition, figure 4 and 6 are somewhat overlapped. Please arrange figures not to show duplicated data.
Response: We believe both MDA-MB-361 and BT474 models are necessary for the study because these two models are established models of HER2+ breast cancer with known differences in HER2 levels. Additionally, this allows for both intergroup and intragroup comparisons. Please note that comparisons in three mouse models showed significant correlations (Figure 4). We agree that part of figure 6 is redundant, and therefore, we removed (e) to avoid the possible overlapping as suggested.
References (please note these have also been added to the manuscript)
[1] Shen G, Huang H, Zhang A, Zhao T, Hu S, Cheng L, et al. In vivo activity of novel anti-ErbB2 antibody chA21 alone and with Paclitaxel or Trastuzumab in breast and ovarian cancer xenograft models. Cancer Immunol Immunother 2011;60(3):339-48.
[2] Ueno NT, Bartholomeusz C, Xia W, Anklesaria P, Bruckheimer EM, Mebel E, et al. Systemic gene therapy in human xenograft tumor models by liposomal delivery of the E1A gene. Cancer Res 2002;62(22):6712-6.
[3] Wojnarowicz PM, Escolano MG, Huang Y-H, Desai B, Chin Y, Shah R, et al. Anti-tumor effects of an Id antagonist with no acquired resistance. bioRxiv 2020:2020.01.06.894840.
[4] Surov A, Meyer HJ, Wienke A. Associations Between PET Parameters and Expression of Ki-67 in Breast Cancer. Transl Oncol 2019;12(2):375-80.

Reviewer 2 Report
I thank Editor and the authors for the opportunity of reviewing such an interesting manuscript.
The authors investigated the potential usefulness of 89Zr-pertuzimab as imaging probe for identifying HER2 expression in vivo and also investigating the relationship between HER2 density, as measured by 89Zr-pertuzumab micro-PET, and response to paclitaxel (PTX) in breast cancer (BC) xenograft mouse model. The authors, utilizing BC xenografts with different levels of HER2 expression (i.e. BT474 with high HER2 density, MDA-MB-361 with low HER2 density, MDA-MB-231 with no HER2 expression) found that HER2 expression is correlated with response to PTX.
I think that it is an excellent research paper, well written and presented, focused on a topic of utmost importance.
Images are illustrative, the scheme representing the experimental work-flow is particularly useful.
Minor considerations:
- the authors chose SUVmean < or > 3 for dichotomizing HER2 expression (low vs high density). I was unable to find in the manuscript the explanation of the reason why the authors chose this threshold: was it the median value? Please explain.
- the authors might add a brief paragraph in the Discussion on the potential usefulness of PET with 89Zr-pertuzmab for the dynamic evaluation also of the acquired resistance in BC patients undergoing PTX.
- few typo errors.
Author Response
Dear Editor and Reviewer,
Thank you for the opportunity to submit a revised manuscript and for the time you have committed to improving the study. We appreciate your efforts and believe we have addressed all of your points as indicated below and in the revised manuscript.
Reviewer #2:
The authors investigated the potential usefulness of 89Zr-pertuzimab as imaging probe for identifying HER2 expression in vivo and also investigating the relationship between HER2 density, as measured by 89Zr-pertuzumab micro-PET, and response to paclitaxel (PTX) in breast cancer (BC) xenograft mouse model. The authors, utilizing BC xenografts with different levels of HER2 expression (i.e. BT474 with high HER2 density, MDA-MB-361 with low HER2 density, MDA-MB-231 with no HER2 expression) found that HER2 expression is correlated with response to PTX.
I think that it is an excellent research paper, well written and presented, focused on a topic of utmost importance.
Images are illustrative, the scheme representing the experimental work-flow is particularly useful.
Response: Thank you for your comments. We appreciate the favorable review and have addressed your minor considerations in the points below.
Minor Considerations:
- The authors chose SUVmean < or > 3 for dichotomizing HER2 expression (low vs high density). I was unable to find in the manuscript the explanation of the reason why the authors chose this threshold: was it the median value? Please explain.
Response: Thank you for your comment. The mean value of [89Zr]-pertuzumab SUVmean was 2.4. There was no value between 2.4 and 3; therefore, an SUVmean < or > 3 was used for dichotomizing HER2 expression as an integer value. However, for clarity, we have changed the thresholding from 3 to 2.4. “By thresholding [89Zr]-pertuzumab at an SUVmean of 2.4 (the mean value of [89Zr]-pertuzumab SUVmean), tumors were classified as high HER2 (SUVmean 2.4), and low HER2 (SUVmean < 2.4) tumors.”
- The authors might add a brief paragraph in the Discussion on the potential usefulness of PET with 89Zr-pertuzmab for the dynamic evaluation also of the acquired resistance in BC patients undergoing PTX.
Response: We appreciate your comment. We have added some brief discussion on this in the “Discussion” (line 257-258).
- Few typo errors.
Response: Thank you for your comment. We have corrected the errors.
Round 2
Reviewer 1 Report
Dear authors
Thank you for your response to my comments.
After reading the revised manuscript, I don't feel that the answer for the major comment 1 is acceptable. The growth rate of tumors is different between human and mice. Answers for the major comment 3 and the minor comment 5 are also partly acceptable.
I appreciate your great efforts on the revision of manuscript.
Sincerely,